# Overexpression of VEGF in the MOPC 315 Plasmacytoma Induces Tumor Immunity in Mice

**DOI:** 10.3390/ijms23095235

**Published:** 2022-05-07

**Authors:** Byung-Gyu Kim, Sung Hee Choi, John J. Letterio, Jie-Young Song, Alex Y. Huang

**Affiliations:** 1Case Comprehensive Cancer Center, Case Western Reserve University School of Medicine, Cleveland, OH 44106, USA; sxc224@case.edu (S.H.C.); john.letterio@uhhospitals.org (J.J.L.); ayh3@case.edu (A.Y.H.); 2Department of Pediatrics, Case Western Reserve University School of Medicine, Cleveland, OH 44106, USA; 3Center for Pediatric Immunotherapy, Angie Fowler AYA Cancer Institute, UH Rainbow Babies & Children’s Hospital, Cleveland, OH 44106, USA; 4Division of Applied Radiation Bioscience, Korea Institute of Radiological & Medical Sciences, Seoul 01812, Korea; immu@kirams.re.kr

**Keywords:** VEGF, cytotoxic T lymphocyte, plasmacytoma, CCR3

## Abstract

Vascular endothelial growth factor (VEGF) has important effects on hematopoietic and immune cells. A link between VEGF expression, tumor progression, and metastasis has been established in various solid tumors; however, the impact of VEGF expression by hematopoietic neoplasias remains unclear. Here, we investigated the role of VEGF in plasma cell neoplasia. Overexpression of VEGF in MOPC 315 tumor cells (MOPCSVm) had no effect on their growth in vitro. However, constitutive ectopic expression of VEGF dramatically reduced tumorigenicity of MOPC 315 when implanted subcutaneously into BALB/c mice. Mice implanted with MOPCSVm effectively rejected tumor grafts and showed strong cytotoxic T lymphocyte (CTL) activity against parental MOPC 315 cells. MOPCSVm implants were not rejected in nude mice, suggesting the process is T-cell-dependent. Adoptive transfer of splenocytes from recipients inoculated with MOPCSVm cells conferred immunity to naïve BALB/c mice, and mice surviving inoculation with MOPCSVm rejected the parental MOPC 315 tumor cells following a second inoculation. Immunohistochemical analysis showed that MOPCSVm induced a massive infiltration of CD3^+^ cells and MHC class II^+^ cells in vivo. In addition, exogenous VEGF induced the expression of CCR3 in T cells in vitro. Together, these data are the first to demonstrate that overexpression of VEGF in plasmacytoma inhibits tumor growth and enhances T-cell-mediated antitumor immune response.

## 1. Introduction

Vascular endothelial growth factor (VEGF), also termed VEGF-A, is a glycosylated, multifunctional cytokine that is abundantly expressed and secreted by most human and animal tumors examined thus far [1,2,3,4]. It increases microvessel permeability approximately 50,000 times more than histamine on a molar basis and stimulates the growth of vascular endothelial cells through binding to its identified receptors, VEGFR-1 (flt-1) and VEGFR-2 (KDR/flk-1), on the surface of endothelial cells [4,5]. Human VEGF is a homodimeric 36–46 kDa protein of four isoforms generated by alternate splicing mechanisms [6]. The 121 and 165 amino acid isoforms are secreted, whereas the 189 and 206 amino acid isoforms appear to remain cell associated [3,4,7]. Mouse VEGF gene exists as three isoforms with 120, 164, and 188 amino acids [8].

VEGF is abundantly expressed in the vast majority of human tumors [9]. They include lung, thyroid, breast, gastrointestinal tract, kidney and bladder, and ovary and uterine cervix carcinoma; angiosarcoma; germ cell tumors; and several intracranial tumors such as glioblastoma multiforme, as well as the VHL syndrome-associated capillary hemangioblastoma [3,4,10]. In solid tumors, VEGF has an important role in the induction of neovascularization correlated with tumor growth and metastatic potential [4,10,11]. Studies also suggest a role of VEGF in hematological malignancies including multiple myeloma (MM), leukemias, and myelofibrosis [12,13,14,15,16,17,18]. VEGF and its receptors are expressed by myeloma [12,16,17], and it is also expressed in plasmacytoma [19,20]. However, the role of VEGF in the pathogenesis of plasmacytomas is unclear.

We hypothesized that the production of VEGF by tumor cells might impair the ability of the immune system to produce an effective and sustained anti-tumor response. Dendritic cells (DCs) are the most potent antigen presenting cells (APCs), and several studies have described the defective function of DCs in tumor-bearing mice and in cancer patients [21,22]. VEGF has been found to inhibit both the maturation and function of DCs in vitro and in vivo [23,24,25]. However, VEGF also induces endothelial cells and promotes monocyte migration via the VEGF receptor flt-1 [26,27,28]. Peripheral blood T lymphocytes and lymphocytes infiltrating human cancers express VEGF [29]. Activation of T cells induces VEGF expression, resulting in induction of its biological activities [30,31]. VEGF has been recently shown to cause thymic atrophy and the inhibition of T cell development and to modulate expression of inhibitory checkpoints on CD8 T cells in tumors [32,33]. On the other hand, VEGF induces Th1 polarization, an enhanced IFN-γ and IL-2 production, and migratory responses in human CD45RO^+^ CD4^+^ memory T cells [34,35]. T cells express the two tyrosine kinases of the VEGF receptor family, Flt-1 (VEGFR-1) and Flk-1 (VEGFR-2), as well as VEGF [34,36]. A recent study showed that VEGF deletion in CD8^+^ T cells leads to enhanced tumor growth and that VEGF-deficient CD8^+^ T cells do not infiltrate tumors efficiently [37]. However, the effect of VEGF on T-cell-mediated tumor immunity remains unclear. To elucidate the immunoregulatory role of VEGF expressed by plasmacytoma cells, we developed a model of the MOPC 315 plasmacytoma that constitutively expresses VEGF and examined the effect of VEGF on the growth of MOPC 315 in vitro and in vivo, as well as on the development of an effective host anti-tumor immune response.

## 2. Results

### 2.1. Ectopic Expression of VEGF Had No Effect on Plasmacytoma Cell Growth In Vitro

To construct the plasmid expressing VEGF, we performed RT-PCR using primers that flank 5′ and 3′ ends of the VEGF cDNA coding region. Similar to previous reports, the mouse plasmacytoma cell line MOPC 315 expressed high levels of VEGF mRNA, including three of the four VEGF splice variants, VEGF120, VEGF164, and VEGF188 (Figure 1A). Since the VEGF164 variant is secreted and is the most effective angiogenic factor in the VEGF family, we assembled both antisense and sense VEGF164 constructs and transfected each individually into MOPC 315 cells, in addition to cells transfected with an empty vector control (Figure 1B). ELISA was performed to analyze the amounts of VEGF proteins secreted in medium supernatant by 1 × 10^5^ cells cultured for 48 h. Whereas the MOPC 315 cells transfected with pEF vector control (MOPCpEF) secreted VEGF at concentrations of 0.7 to 0.9 ng/mL (100%), the cells transfected with antisense VEGF genes (MOPCASVm) showed a significant reduction in VEGF production (30% less than control cells). Cells transfected with sense VEGF genes (MOPCSVm) produced VEGF at 20- to 24-fold higher levels (Figure 1C). To determine whether the change of VEGF expression in the transfectants was the indirect consequence of an autocrine effect on cell growth, the MOPCpEF, MOPCASVm, or MOPCSVm were each seeded as six replicates at two different densities, and their growth rates were monitored over 2 days. As seen in Figure 1D, growth rates of these cells were almost identical, indicating that the observed change in VEGF expression was not secondary to changes in cell number.

### 2.2. Reduced Tumorigenicity of VEGF-Expressing MOPC 315 Transfectants in BALB/c Mice

The MOPCpEF, MOPCASVm, or MOPCSVm cells (1 × 10^6^ cells/mice) were injected subcutaneously into BALB/c mice. Tumors failed to progress in 5/10 mice injected with MOPCSVm during a 25-day period in which progressive growth of MOPCpEF or MOPCASVm cells was invariably observed (Figure 2). Tumors developed in all 10 mice challenged with MOPCpEF cells (Figure 2A), as well as in 8 out of 10 mice challenged with MOPCASVm (Figure 2B). Furthermore, in the recipients of MOPCSVm cells in which tumors developed, there was a notable delay in progression (Figure 2C), with tumors typically appearing 5 days later than those developing in the recipients of either MOPCpEF or MOPCASVm cells. The tumor volume of MOPCSVm was also smaller, compared with either MOPCpEF or MOPCASVm (Figure 2D). The tumor incidence of MOPCSVm decreased significantly to 50% (Figure 2E).

### 2.3. Development of Cell-Mediated Immunity in Mice Implanted with MOPCSVm Cells

We hypothesized that the reduction in tumor growth and progression in mice implanted with MOPCSVm cells was due to induction of a host anti-tumor immune response. To evaluate whether VEGF expression might modulate the antigen-presenting capacity of tumor cells, we first checked the cell surface expression of MHC class I, class II, B7.1, and ICAM-1 on transfectants by FACS analysis and found no difference between transfectants and control cells (data not shown). To determine whether the in vivo challenge with either MOPCpEF or MOPCSVm induced a tumor-specific T-cell response, splenocytes were isolated from mice in each group, stimulated in vitro with mitomycin-C-treated MOPC 315 cells, and then assayed for their ability to kill MOPC 315 target cells in a standard chromium release assay. As shown in Figure 3A, effector T cells from the mice challenged with MOPCSVm exhibited higher specific lysis than cells from the mice challenged with MOPCpEF, whether in the absence or presence of IL-2 (10 U/mL). Cytolytic activity of the cytotoxic T lymphocytes (CTLs) against the MOPC 315 cells was significantly blocked by pretreatment of the target cells with anti H-2 K^d^D^d^ mAb (Figure 3B). This result indicates that VEGF overexpressed from MOPCSVm can enhance the activity of CTL.

### 2.4. Regression of MOPCSVm Tumors Was T-Cell-Dependent

To directly demonstrate a role for T cells in the regression of MOPCSVm tumors in vivo, MOPC 315 plasmacytoma syngeneic immunocompetent BALB/c mice and immunocompromised BALB/c nude mice (T cell deficient) were simultaneously transplanted *s.c.* with 1 × 10^6^ MOPCpEF or MOPCSVm cells (Figure 4A). While the growth rates of MOPCSVm in syngeneic immunocompetent BALB/c mice were decreased relative to controls, their progression in T-cell-deficient BALB/c nude mice was increased when compared to the growth of MOPCpEF control cells. Both MOPCpEF and MOPCSVm tumors grew more rapidly in the BALB/c nude recipients than in BALB/c mice (Figure 4A). These data demonstrate the T cell dependency in the regression of tumors. To determine whether the MOPCSVm-induced immunity could be passively transferred to naïve BALB/c mice, non-adherent spleen T cells from mice previously challenged with MOPCpEF or MOPCSVm for 18 days were transferred into the lethally irradiated mice. All mice were implanted *s.c.* with 1 × 10^6^ parental MOPC 315 cells 24 h after injection of splenocytes. As shown in Figure 4B, the non-adherent spleen cells from mice challenged with MOPCSVm were highly effective in retarding tumor growth when compared to splenocytes of mice challenged with MOPCpEF. These results demonstrate the establishment of T cell memory in recipients of MOPCSVm cells. To further document MOPC 315-specific T-cell memory, parental MOPC 315 cells were inoculated either in mice pre-challenged with MOPCSVm (Figure 4D) or in previously unchallenged control mice (Figure 4C). Tumor volume in mice pre-challenged with MOPCSVm was significantly smaller when compared to the control mice (Figure 4C,D). Tumor incidence in mice pre-challenged with MOPCSVm was also decreased by 80% when compared to the control mice (Figure 4 4C,D and Appendix A). Taken together, these data demonstrate that MOPC 315 plasmacytoma overexpressing VEGF can induce anti-tumor immunity, such as effector CTL memory.

### 2.5. Diffuse Infiltration of Leukocytes around MOPCSVm

Next, we performed an analysis to examine the microenvironment of tumor sites in mice implanted with MOPCpEF or MOPCSVm (Figure 5). In hematoxylin-and-eosin-stained sections, tumor tissues from mice challenged with MOPCSVm (Figure 5B) contained a more extensive infiltration of leukocytes as compared to sites of MOPCpEF inoculation (Figure 5A). Immunohistochemical analysis for T cells (anti-CD3, Figure 5C–F) showed a stronger and diffuse infiltration of CD3^+^ T cells in the recipients of MOPCSVm (Figure 5D,F) compared to that in the recipients of MOPCpEF (Figure 5C,E). These data were confirmed with flow cytometry analysis (Appendix A). In addition to CD3^+^ T cells, IHC analysis showed more infiltration of CD8^+^ T cells in the recipients of MOPCSVm (Appendix A). There was also a significant increase in the number of MHC class II^+^ antigen-presenting cells (APCs) in tissue sections from recipients of MOPCSVm cells (Figure 5H,J), compared to that in the recipients of MOPCpEF (Figure 5G,I).

### 2.6. Induction of Chemokine Receptor Expression and Angiogenesis by VEGF

To evaluate mechanisms underlying the increased infiltration of T cells into MOPCSVm tumors, we measured the expression of chemokine receptors on T cells exposed to exogenous VEGF. Purified lymph node T cells were stimulated with anti-CD3 and anti-CD28 Abs in absence or presence of VEGF. After 24 h, we isolated mRNA from each group for RT-PCR. VEGF-treated T cells expressed CCR3 at a 4.8-fold higher level (Figure 6A), while both T cell populations expressed little CCR5 (data not shown). We also analyzed angiogenesis in vivo by microvascular density (MVD). Tumors grown from MOPCSVm were associated with more prominent microvasculature compared with MOPCpEF (Figure 6B). MVD in tumors generated by MOPCSVm cells was significantly higher than that observed in control tumors, as determined by quantitative analysis of CD31 staining density.

## 3. Discussion

VEGF is a potent stimulator of angiogenesis in vivo, and the important role of this cytokine in tumor biology is suggested by the observation that VEGF is commonly upregulated in solid tumors [3,4,38]. VEGF production is also a feature of human myeloma cells [12,16,17,18] and murine plasmacytomas [19]; however, the role of VEGF in the in vivo progression of plasma cell tumors is not well defined. In our current study, constitutive expression of either antisense or sense VEGF genes in the murine MOPC 315 plasmacytoma tumor cell line was utilized as a means to elucidate the effect of VEGF in the growth of plasmacytoma cells in vivo, as well as to directly address the possible application of antisense or sense VEGF gene therapy in the treatment of multiple myeloma. MOPC 315 plasmacytoma cells were found to express and secrete three of the four VEGF splice variants, VEGF120, VEGF164, and VEGF188 (Figure 1A), consistent with the findings reported by Paydas et al. [19]. Surprisingly, the in vivo progression of MOPC 315 plasmacytoma cells transfected with sense VEGF genes (MOPCSVm) in BALB/c mice was inhibited when compared with MOPC 315 cells transfected with either a control vector (MOPCpEF) or with antisense VEGF genes (MOPCASVm) (Figure 2).

VEGF has been shown to induce tumor angiogenesis, increase tumor growth and metastasis, and inhibit maturation and function of DCs [3,4,23,24,25], all properties which enhance tumor growth and metastasis. On the other hand, VEGF also has demonstrated activities that may potentiate host anti-tumor immune response. For example, VEGF induces monocyte migration via the VEGF receptor Flt-1 [26,27]. In addition, T cell activation induces expression of VEGF in T cells [30,31], and the VEGF expressed in activated T cells can induce Th1 differentiation, augment interferon-γ production, and inhibit IL-10 secretion [34,35]. It was reported that the immune system is associated with the progression of myeloma [39,40], underlying impaired NK cell cytotoxicity in patients with multiple myeloma [41,42]. However, the effect of VEGF on T-cell-dependent myeloma immunity remains unclear. Our observation that spleen cells from mice challenged with the VEGF-expressing MOPCSVm exhibited higher specific lysis when cultured with mitomycin-C-treated MOPC 315 tumor cells in vitro, suggest a direct role for VEGF in the induction of T cell memory (Figure 3). T cell dependency of MOPCSVm tumor regression in vivo was also confirmed by the more rapid progression of this tumor in T-cell-deficient nude mice. The adoptive transfer of tumor immunity by the non-adherent spleen cells from mice previously challenged with MOPCSVm strongly indicates that the regression of MOPCSVm in BALB/c mice is critically dependent on T cells (Figure 4).

In the current study, we show that VEGF overexpressed from MOPC 315 led to reduced tumor growth by enhanced CTL activity and increased CD8^+^ T infiltration into tumors. Consistent with our observation, Palazon et al. demonstrated the importance of the HIF-1α/VEGF axis in tumor immunity [37]. Genetic deletion of *HIF-1α*, but not *HIF-2α*, in T lymphocytes leads to accelerated tumor growth and impaired CD8^+^ T cell tumor infiltration in Lewis lung carcinoma (LLC) and B16F10 melanoma cells. Deletion of *VEGF*, an HIF target gene, in T cells also accelerates tumorigenesis. As the primary source of VEGF in a tumor will often be the malignant cells themselves, the relationship between the tumor, infiltrating TILs, and the production of VEGF is difficult to place in an etiologic order given their observations. However, Palazon et al. showed the role for VEGF produced in CD8^+^ T cells to inhibit tumor growth by an increase in CD8^+^ T cells migrating into tumors and, in part, by enhanced activity of CTL killing tumor cells. Our findings provide further a rationale for the ongoing clinical evaluation of combinatorial therapies comprising immunotherapies and anti-angiogenic approaches [43,44].

We found that tumor tissues from the mice challenged with MOPCSVm had a more extensive infiltration of leukocytes than tumors from mice challenged with MOPCpEF. Immunohistochemical analysis revealed that these infiltrates mainly consisted of CD3^+^ T cells, CD8^+^ T cells, and MHC class II^+^ APC (Figure 5). The CD3^+^ cells, CD8^+^ cells, and MHC class II^+^ cells were widely dispersed in the MOPCSVm tumor tissue, but were typically localized at the outer margin of MOPCpEF tumors. These data indicate that VEGF expression by MOPCSVm can increase the presence of APC within tumor tissue as a mechanism underlying the enhanced CTL generation. A previous report by Barleon et al. [26] has shown that VEGF promotes monocyte migration via the VEGF receptor Flt-1. As CCR3 is known to be expressed in T cells and involved in CD4^+^ and CD8^+^ T cell trafficking to inflammation and infiltration in the tumors [45,46], the induction of CCR3 expression by VEGF in T cells stimulated with anti-CD3 and anti-CD28 Abs suggests a novel mechanism by which tumor-derived VEGF could induce the migration of T cells into a tumor site, in addition to the direct effects of signals immediately downstream of the VEGFR-1 or VEGFR-2 receptors.

Leukocyte infiltration of tumors could also be indirectly enhanced by VEGF as a consequence of the induction of neovascularization. We analyzed angiogenesis in vivo by microvascular density (MVD). Tumors grown from MOPCSVm were associated with more prominent microvasculature compared with MOPCpEF. In a T-cell-deficient background, the overexpression of VEGF in plasmacytoma cells enhanced tumor growth (Figure 4A), and MVD was significantly higher compared with control tumors (Figure 6B). This observation is consistent with a previous report by Claffey et al. [47] that showed that the expression of VEGF by SK-MEL-2 melanoma cells increases tumor growth and metastasis in a manner associated with induction of angiogenesis. In preliminary experiments, we also observed an increased angiogenesis associated with a more rapid in vivo growth of B16F10 melanoma cells transfected with sense VEGF genes (B16F10SVm) relative to the parental B16F10 tumor cells transfected with a control vector (B16F10pEF) in C57BL/6N mice (Appendix A). In these melanoma models, however, the ectopic expression of VEGF does not induce tumor immunity. Thus, the effect of VEGF on tumor growth in vivo is complex and ultimately determined by the balance between effects on immune cells and on angiogenesis.

In conclusion, decrease in NK cell activity is significantly associated with advanced clinical stage in multiple myeloma patients [41,48]. NK cells can trigger angiogenesis in tumor under specific circumstances, but also their cytotoxic functions can be reduced by VEGF signaling [49,50]. VEGF produced in CD8^+^ T cells inhibits tumor growth by an increase in CD8^+^ T cells migrating into tumors and, in part, by enhanced activity of CTL to kill tumor cells [37]. In the current study, overexpression of VEGF in a model of plasma cell neoplasia enhanced tumor infiltration by T cells and antigen-presenting cells, leading to the induction of tumor-specific immune responses. These results suggest a potential role for VEGF as an immune modulator in the development of plasmacytoma immunotherapies [17,51].

## 4. Materials and Methods

### 4.1. Tumor Cell Lines and Cell Culture Conditions

The MOPC 315 plasmacytoma cell line (American Type Culture Collection, Rockville, MD, USA) was cultured in vitro in Dulbecco’s modified Eagle’s medium (DMEM) containing penicillin, streptomycin, and 10% fetal calf serum (FCS). Female BALB/c and athymic BALB/c/nu/nu (BALB/c nude) mice were purchased from the Charles River Breeding Center (Atsugi, Ayase, Japan) and maintained in a specific pathogen-free condition. To analyze the effect of VEGF on the in vivo growth, MOPC 315 cells were transfected with VEGF genes and then injected subcutaneously into BALB/c mice. The animal experiments were performed in accordance with institutional guidelines and with approval of the Institutional Animal Care and Use Committee at Case Western Reserve University and at the Korea Institute of Radiological and Medical Sciences.

### 4.2. RNA Isolation and RT-PCR

Total RNA was isolated from the MOPC 315 plasmacytoma cells using the Trizol reagent (Gibco BRL, Carlsbad, CA, USA). Reverse transcription was performed with 2 μg of RNA in a 25 μL final volume containing 10 mM Tris-HCl (pH 8.3), 15 mM KCl, 2 mM dithiothreitol, 0.6 mM MgCl2, 2.5 mM deoxynucleotides, 20 U of RNasin (Promega, Madison, WI, USA), 100 U of reverse transcriptase (Promega, Madison, WI, USA), and 100 pmol oligo-dT for 1 h at 37 °C. A portion of the reaction mixture (2 μL) was subjected to PCR amplification in 25 μL of a reaction mixture containing 10 pmol of each primer, 1.5 mM MgCl2, 10 mM Tris-HCl (pH 8.3), 50 mM KCl, 0.2 mM each of four deoxynucleotides, and 0.5 U Taq polymerase (Promega, Madison, WI, USA). The VEGF isoforms were amplified by PCR using primers that flank 5′ and 3′ ends of the VEGF cDNA coding region, respectively, GAAGCGGCCGCAACCATGAACTTTCTGCTCTCTTGG (sense) and GAAGCGGCCGCTCACCGCCTTGGCTTGTCACA (antisense), modified to contain Not I sites. PCR amplification of VEGF cDNA was performed over 30 cycles under the following conditions: 1 min at 94 °C, 1 min at 65 °C, and 1 min at 72 °C. Expression of CCR3 and CCR5 was performed by RT-PCR using a mouse/rat CCR3 and CCR5 Primer Pair (R&D Systems, Minneapolis, MN, USA) according to the manufacturer’s instructions [45].

### 4.3. Construction of VEGF Vector

The mouse VEGF 120 and VEGF 164 cDNA were initially isolated from MOPC 315 cells by RT-PCR and subcloned into the pEF eukaryotic expression vector (Invitrogen, Carlsbad, CA, USA) using the Not I restriction enzyme site. Restriction enzyme analysis (Bsa I) and DNA sequencing were used to confirm the antisense and sense orientations and sequences of the VEGF cDNA in the pEF vector from individual transformants.

### 4.4. Transfection and Quantification of VEGF

The transfection was performed using 2 μg of the VEGF construct DNA by Lipofectamine Plus^TM^ Agent (Invitrogen, Carlsbad, CA, USA). After 48 h or when the cells were confluent, they were replated at a dilution of 1:20 in selection medium containing Neomycin G418 (500 μg/mL). VEGF was detected in MOPC 315 cell supernatants by using an enzyme-linked immunosorbent assay (ELISA) kit specific for murine VEGF (R&D Systems, Minneapolis, MN, USA). Recombinant mouse VEGF, purchased from R&D Co., was used as the standard.

### 4.5. Assay of In Vitro Growth Rate

Transfected cells (vector alone, antisense, or sense VEGF) were cultured (1 × 10^5^ cells/mL) under standard culture conditions. Cells were counted in a hemocytometer by the trypan blue exclusion method, initially at 12 h time points and then every 24 h for a total of at least 48 h. The total number of cells from duplicate experiments was determined as a function of time, and the rate of division was calculated from the exponential phase of growth.

### 4.6. Tumorigenicity and Immunization of Animals

MOPC 315 plasmacytoma cells transfected with control vector (MOPCpEF), antisense VEGF genes (MOPCASVm), or sense VEGF genes (MOPCSVm) were cultured in medium and then subcutaneously injected into BALB/c and BALB/c nude mice. Transfected cells (1 × 10^6^ cells in 100 μL of PBS) were injected subcutaneously into the right flank of BALB/c and BALB/c nude mice at 8–10 weeks of age [52]. Tumors were measured by caliper at three-day intervals and scored six weeks after injection. In addition, parental MOPC 315 cells were injected in the left flank of control (uninoculated) mice and mice previously challenged with MOPCSVm (in whom tumors completely regressed). All experiments were performed four times, with each group consisting of more than 10 mice.

### 4.7. Mixed Lymphocyte Tumor Cell Cultures (MLTC)

For MLTC, 3 × 10^6^ spleen cells from BALB/c mice challenged with MOPCpEF or MOPCSVm were cultured with 2 × 10^5^ mitomycin-C-treated MOPC 315 cells in 24-well culture plates for 5 days [52].

### 4.8. Cytotoxicity Assay

After 5 days in culture, MLTC cells were harvested, and the cytotoxic T lymphocyte (CTL) activity was assayed against ^51^Cr-labeled target cells. To label the target cells, 2 × 10^6^ cells were incubated in 0.2 mL culture medium with 200 μCi Na_2_[^51^Cr]O_4_ (New England Nuclear, Boston, MA, USA) for 60 min at 37 °C. Variable numbers of effector cells were mixed with 1 × 10^4 51^Cr-labeled target cells in triplicate and cultured in 96-well round-bottomed tissue culture plates. After 4 h, the content of the plates was centrifuged at 400× *g* for 5 min, and 0.1 mL of the supernatant was counted in a gamma counter. Spontaneous ^51^Cr release was determined by incubating the target cells with medium alone [52]. The percentage of specific release was calculated as
$$\mathrm{Specific}\;\mathrm{release}\;(\%)=100\;\times \;\frac{\left(51_{Cr\;\mathrm{experimental}\;\mathrm{release}}\right)-\left(51_{\mathrm{Cr}\;\mathrm{spontaneous}\;\mathrm{release}}\right)}{\left(51_{\mathrm{Cr}\;\mathrm{maximun}\;\mathrm{release}}\right)-\left(51_{\mathrm{Cr}\;\mathrm{spontaneous}\;\mathrm{release}}\right)}$$

Maximum release was determined by lysis of labeled target cells with a detergent.

### 4.9. Neutralization Test

For neutralization test, target cells were pretreated with anti-(H-2 K^d^D^d^) mAb (IgM, Cedarlane, Burlington, NC, USA). For the isotype-matched controls, the target cells were pretreated with purified anti-TNP (trinitrophenol) Ab (IgM, BD Pharmingen, Franklin Lakes, NJ, USA).

### 4.10. Adoptive Transfer

The immune cells of the syngeneic recipient are inactivated by exposing the host to irradiation (800 rad) (CGRmev, Munich, Germany). The lethally irradiated syngeneic mice were intravenously injected with 1 × 10^6^ non-adherent spleen cells from MOPCpEF-primed mice or MOPCSVm-primed mice on the day following irradiation and were then subcutaneously implanted with 1 × 10^6^ parental MOPC 315 cells on the third day after irradiation [53].

### 4.11. Immunohistochemical Analysis

Tissues from animals receiving either MOPCpEF or MOPCSVm cells were snap-frozen in O.C.T. compound and stored at −80 °C. Then, 6 μm tissue sections were air-dried and fixed in acetone, rehydrated in PBS, and blocked with 2% normal goat serum in PBS for nonspecific staining. Primary anti-mouse CD3, CD8, H-2 I-A^d^/I-E^d^, or CD31 (BD PharMingen, Franklin Lakes, NJ, USA) were applied for 60 min at room temperature. All slides were subsequently washed in PBS and incubated for 30 min with biotinylated secondary Abs (goat anti-rat IgG). Bound Abs were then detected with ABC reagent and Vectastain substrate kit (DAKO, Santa Clara, CA, USA) [54]. For the hematoxylin and eosin staining, murine tissues were fixed in 10% neutral buffered formalin and embedded in paraffin. Then, 4 μm sections were cut and stained with hematoxylin and eosin.

### 4.12. Statistical Analysis

In vitro experiments were performed in triplicate and independently at least three times. In vivo experiments were performed with groups of 10 mice. The significance of differences was determined by a two-sample Student’s *t*-test. Statistical significance was accepted to be a *p*-value less than or equal to 0.05, with * *p* < 0.05, ** *p* < 0.01, *** *p* < 0.001.

## Figures and Tables

**Figure 1 ijms-23-05235-f001:**
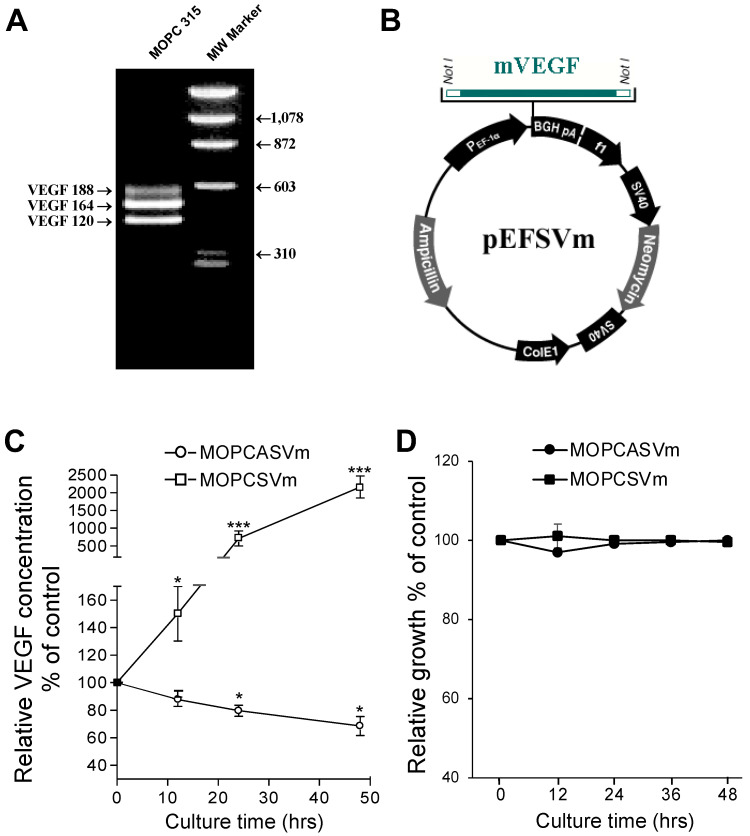
Absence of autocrine VEGF effect on in vitro growth of MOPC 315 cells. (**A**) Alternate splicing of VEGF/RT-PCR analysis. Total RNA isolated from the MOPC 315 cells was used for expression by RT-PCR. Results of RT-PCR show that VEGF 120, VEGF 164, and VEGF 188 were the most abundant isoforms expressed in the mouse plasmacytoma cell line. Molecular weight markers are shown at the right margin. The VEGF isoforms indicated the positions of expressed VEGF. (**B**) Construction of the expression vector for VEGF. Not I sites were created at the 5′ and 3′ ends of cDNA coding for VEGF. The modified cDNA was inserted into the vector pEF downstream from the PEF-1α promotor. (**C**) Stable transfection and expression of sense and antisense VEGF genes. Transfection was performed as described under the Materials and Methods section. ELISA was performed to analyze the amounts of VEGF proteins secreted in medium supernatant after 1 × 10^5^ cells were cultured for 48 h. (**D**) In vitro growth rate of the transfected cells. The transfected cells were counted in a hemocytometer by the trypan blue exclusion method at 12 h, 24 h, and 48 h time points. * *p* < 0.05, *** *p* < 0.001.

**Figure 2 ijms-23-05235-f002:**
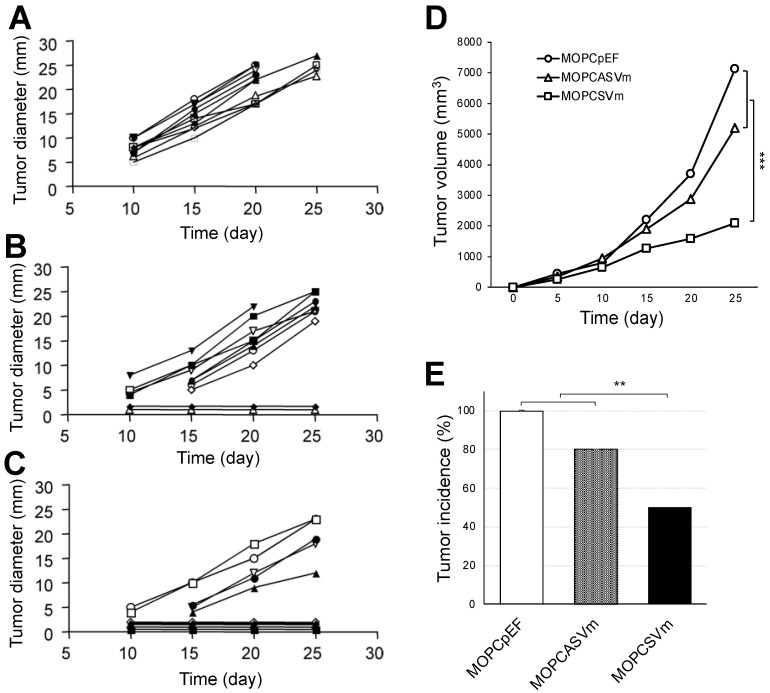
Effect of VEGF on the growth and regression pattern of MOPC 315 cells. BALB/c mice at 8–10 weeks of age were transplanted subcutaneously with 1 × 10^6^ MOPCpEF (**A**), MOPCASVm (**B**), or MOPCSVm (**C**). Each group consisted of 10 mice, and each line refers to a single mouse. The tumor incidences in mice transplanted with MOPCpEF, MOPCASVm, and MOPCSVm were 100%, 80%, and 50%, respectively. (**D**) Tumor volume. (**E**) Tumor incidence. Tumors were measured by caliper, and MOPCSVm tumors appeared 5 days later than the MOPCpEF and MOPCASVm tumors. All experiments were performed three times. ** *p* < 0.01, *** *p* < 0.001.

**Figure 3 ijms-23-05235-f003:**
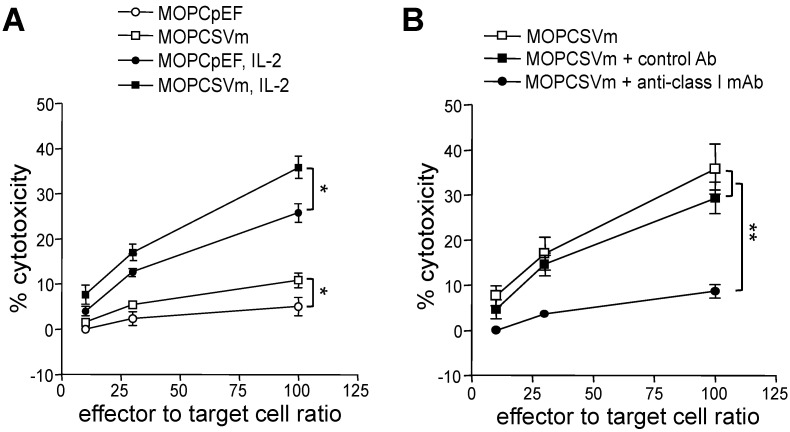
Generation of cytotoxic effector cells in mice implanted with MOPCSVm. Splenocytes obtained from the mice challenged with 1 × 10^6^ MOPCSVm or with 1 × 10^6^ MOPCpEF were stimulated with mitomycin-C-treated MOPC 315 cells for 5 days in the absence or presence of IL-2 (10 U/mL), and their ability to kill MOPC 315 target cells was tested in a standard chromium release assay. (**A**) Effector cells from the mice challenged with MOPCSVm displayed higher cytotoxic lysis of MOPC 315 target cells than those from the mice challenged with MOPCpEF in the absence and presence of IL-2. Error bars represent the +SD of triplicates. These results are representative of five independent experiments. (**B**) To test whether the cytotoxic effector cells activity observed was restricted to the MHC class I antigen, the target cells were treated with the anti H-2K^d^D^d^ mAb. As the negative controls, targets cells pretreated with isotype-matched control antibody were used in the same assay. * *p* < 0.05, ** *p* < 0.01.

**Figure 4 ijms-23-05235-f004:**
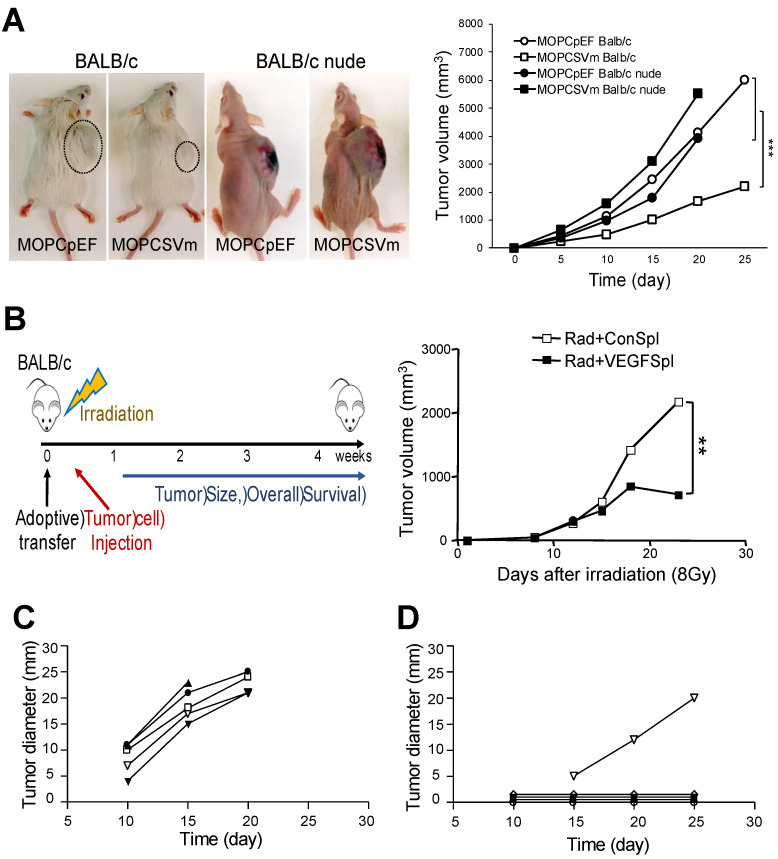
T cell dependency of the MOPCSVm rejection. (**A**) Groups of BALB/c mice and BALB/c nude mice were simultaneously transplanted *s.c.* with either 1 × 10^6^ MOPCpEF cells or MOPCSVm cells. The growth rate of MOPCSVm was decreased in BALB/c mice but was slightly increased in nude mice when compared to growth of MOPCpEF control cells. (**B**) Adoptive transfer of non-adherent spleen cells from MOPCSVm-primed mice suppressed tumor growth. Scheme of animal experiments (left). Syngeneic recipient mice were irradiated and reconstituted with the non-adherent spleen cells prepared from either MOPCpEF-primed mice (open squares) or from MOPCSVm-primed mice (closed squares) and implanted *s.c.* with the parental tumor cells (right). (**C**,**D**) Memory of the CTL response in mice pre-challenged with MOPCSVm was demonstrated by inoculating the parental MOPC 315 cells into naïve mice (**C**) or into recipients of MOPCSVm cells in which tumors had regressed (**D**). ** *p* < 0.01, *** *p* < 0.001.

**Figure 5 ijms-23-05235-f005:**
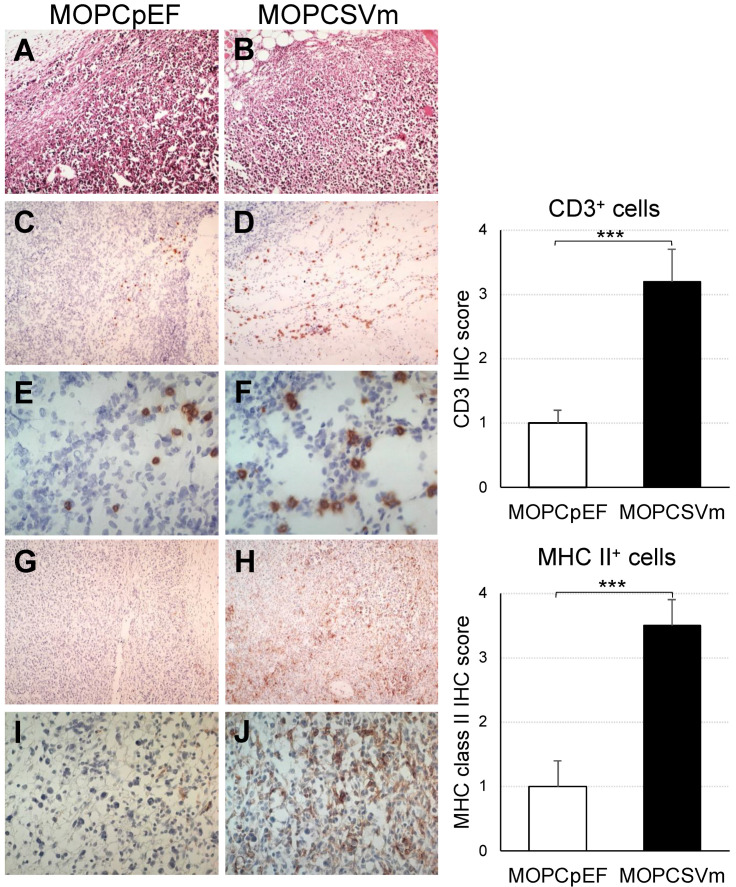
Diffuse infiltration of leukocytes in MOPCSVm tumor sites. (**A**,**B**) Photomicrographs of hematoxylin-and-eosin-stained tissue sections of tumor sites. Tumor tissues from the recipients of MOPCpEF and MOPCSVm were processed and stained with hematoxylin and eosin. Tumor tissue from the mice transplanted with MOPCSVm (**B**) exhibited stronger infiltration of leukocytes than that from the mice transplanted with MOPCpEF (**A**). Immunohistochemical staining with anti-CD3 mAb (**C**–**F**) and with anti-MHC class II mAb (**G**–**J**) in tumor sites. Tumor tissue from the mice transplanted with MOPCSVm (**D**,**F**) exhibited stronger infiltration of CD3^+^ cells than that from the mice transplanted with MOPCpEF (**C**,**E**). Tumor tissue from the mice transplanted with MOPCSVm (**H**, **J**) exhibited diffuse infiltration of anti-MHC class II^+^ cells, but tumor tissue from the mice transplanted with MOPCpEF (**G**,**I**) exhibited local infiltration (original magnification ×100, (**A**–**D**,**G**,**H**); ×400, (**E**,**F**,**I**,**J**)). Data represent average ± S.E. (n = 9). *** *p* < 0.001.

**Figure 6 ijms-23-05235-f006:**
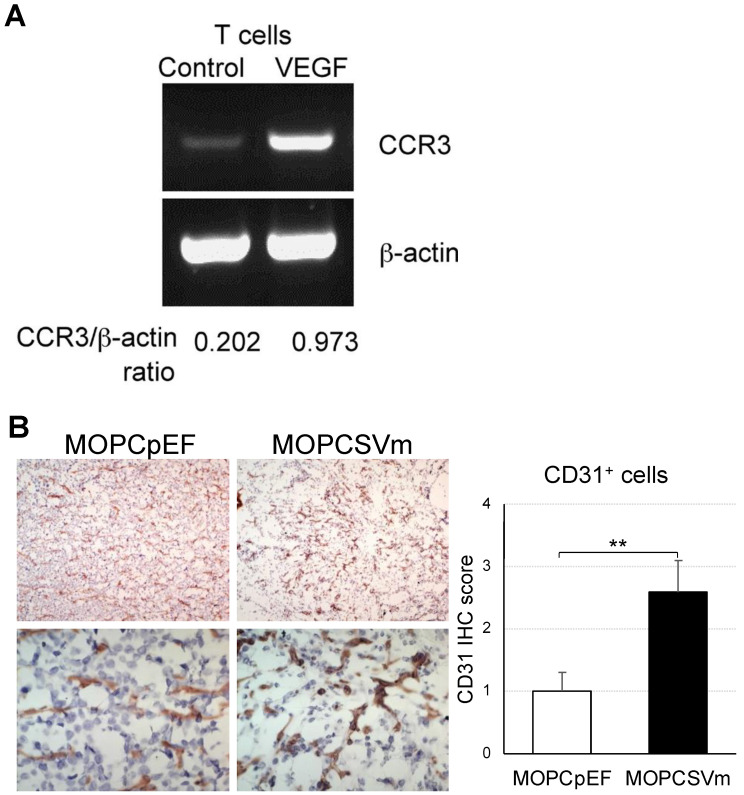
Effect of VEGF expression on chemokine receptor expression in T cells and on tumor-related angiogenesis. (**A**) RT-PCR analysis for CCR3 expression in T cells. T cells stimulated by 1 μg/mL of anti-CD3 and 0.5 μg/mL of anti-CD28 were incubated in absence or presence of VEGF (15 ng/mL) for 24 h. (**B**) Immunohistochemical staining with anti-CD31 mAb in tumor sites. Tumor tissue from the mice transplanted with MOPCSVm exhibited strong staining of anti-CD31^+^ cells when compared to tissues of mice transplanted with MOPCpEF (original magnification ×100, top panels; ×400, bottom panels). Error bars indicate S.E.; ** *p* < 0.01 compared with MOPCpEF.

## Data Availability

The data presented in this study are available in article and Appendix A.

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
