# Peer review of "Overexpression of VEGF in the MOPC 315 Plasmacytoma Induces Tumor Immunity in Mice"

_ijms, 2022, doi:10.3390/ijms23095235_

Round 1

Reviewer 1 Report

The study showing that tumor growth is inhibited in MOPCSVm, a plasma cell type that strongly expresses VEGF, and that this is T cell-dependent is exciting data as it relates to VEGF.

1. In Figure 5, the increase of tumor-infiltrating lymphocytes should be assessed directly by flow cytometry, as immunostaining is relatively poorly quantitative.  It should be shown in a highly quantitative manner.
2.  In the discussion part of l388, you wrote, " We found that tumor tissues from mice challenged with MOPCSVm have a more 388 extensive infiltration of leukocytes than tumors from mice challenged with MOPCpEF.389 Immunohistochemical analysis revealed that these infiltrates mainly consisted of CD3+ T 390 cells, CD8+ T cells and MHC class II+ APCs (Figure 5)." The rationale for mentioning CD8+T cells infiltration is unclear. Please clearly state the reason for this.

Author Response

The study showing that tumor growth is inhibited in MOPCSVm, a plasma cell type that strongly expresses VEGF, and that this is T cell-dependent is exciting data as it relates to VEGF.

Point 1: In Figure 5, the increase of tumor-infiltrating lymphocytes should be assessed directly by flow cytometry, as immunostaining is relatively poorly quantitative.  It should be shown in a highly quantitative manner.

Response 1: We appreciate this point. We have provided the flow cytometry analysis in Supplementary Figure 2A that shows an increase in CD3+ T cells in the MOPCSVm tumor sites, compared to those in the MOPCpEF tumor sites.

Point 2: In the discussion part of l388, you wrote, " We found that tumor tissues from mice challenged with MOPCSVm have a more 388 extensive infiltration of leukocytes than tumors from mice challenged with MOPCpEF.389 Immunohistochemical analysis revealed that these infiltrates mainly consisted of CD3+ T 390 cells, CD8+ T cells and MHC class II+ APCs (Figure 5)." The rationale for mentioning CD8+T cells infiltration is unclear. Please clearly state the reason for this.

Response 2: We apologize for not showing the rationale for mentioning CD8+ T cell infiltration. We have added the IHC staining analyses of CD8+ T cells in Supplementary Figure 2B that show an increase in MOPCSVm tumor sites, compared to those in MOPCpEF tumor sites.

Reviewer 2 Report

the work has been corrected

Author Response

We appreciate the favorable comments from Reviewer.

This manuscript is a resubmission of an earlier submission. The following is a list of the peer review reports and author responses from that submission.

Round 1

Reviewer 1 Report

In this manuscript, Byung-Gyu Kim et al., showed that VEGF-a overexpression in tumor cells increases T cell infiltration and decreases tumor growth. It is interesting manuscript asking important question but there are major concerns that need to be answered:

1- There is no statistical analysis performed on the experiments and that make it difficult to interpret data

2- VEGF did not change tumor growth in Figure 1 but in Figure 4 tumor growth is obviously increased in mice received VEGF positive tumor cell line. Needs clarification.

3- All in vivo tumor growth data should be presented as tumor volume and appropriate statistical analysis should be applied.

4- It is not clear why authors used IL-2 in Figure 3? IL-2 is not known for cytotoxic function.

5- The expression of MHC-II does not mean that there is infiltration of professional APCs as epithelial cell and fibroblasts can express PHC-II as well.

Minor:

Figure 1 is very confusing. Authors can separate tumor cell growth from VEGF production and make two separate graphs.

Author Response

Comments from the Editors and Reviewers:

Reviewer #1: In this manuscript, Byung-Gyu Kim et al., showed that VEGF-a overexpression in tumor cells increases T cell infiltration and decreases tumor growth. It is interesting manuscript asking important question but there are major concerns that need to be answered:

1- There is no statistical analysis performed on the experiments and that make it difficult to interpret data

We appreciate this point by Reviewer 1. We have added the statistical analysis in each figure needed.

2- VEGF did not change tumor growth in Figure 1 but in Figure 4 tumor growth is obviously increased in mice received VEGF positive tumor cell line. Needs clarification.

We appreciate these inquiries. There is a difference in the tumor growth between Figure 1 and Figure 4. While the tumor growth in Figure 1C is in vitro cell growth measured by trypan blue exclusion assay, the tumor growth in Figure 4A is in syngeneic immunocompetent BALB/c mice (Left in Fig. 4A) and in syngeneic immunocompromised BALB/c nude mice (Right in Fig. 4A). Figure 1C shows no tumor intrinsic effect of VEGF in MOPC 315 cell growth. Figure 4A shows the tumor extrinsic effect of VEGF in tumor microenvironment as a decrease in tumor growth in BALB/c mice is due to the CD8 cytotoxic T cells and an increase in BALB/c nude (T cell deficient mice) is due to the increased tumor angiogenesis.

3- All in vivo tumor growth data should be presented as tumor volume and appropriate statistical analysis should be applied.

We appreciate this suggestion. The previous in vivo tumor growth data showed the tumor incidence as well as tumor size. We have added tumor volume for the in vivo tumor growth in Supplementary Figure 1 supporting Figure 2 and in Supplementary Figure 2 for Figure 4A.

4- It is not clear why authors used IL-2 in Figure 3? IL-2 is not known for cytotoxic function.

We appreciated this inquiry. IL-2 was initially called T cell growth factor. It stimulates proliferation and enhances function of T cells. Therefore, we used IL-2 in this CTL assay to promote CD8 T cell expansion and effector differentiation as IL-2 signals are also necessary for memory response [1-3]. As shown in Figure 3, IL-2 addition enhanced the cytotoxic activity without influencing the cytotoxicity patterns.

5- The expression of MHC-II does not mean that there is infiltration of professional APCs as epithelial cell and fibroblasts can express PHC-II as well.

We appreciate this point. Although MHC class II molecules can be expressed on epithelial cell and fibroblasts as the reviewer mentioned [4], these molecules are selectively expressed on antigen-presenting cells including dendritic cells, macrophages, and B cells [5]. As shown in Figure 5, we confirmed with pathologists that the cells expressing MHC II molecules are APCs, not epithelial cell or fibroblasts.

Minor:

Figure 1 is very confusing. Authors can separate tumor cell growth from VEGF production and make two separate graphs.

We appreciate this suggestion. We have now made two separate graphs as Figures 1C and 1D.

Reviewer 2 Report

In this study, tumor growth was suppressed in MOPCSVm, a plasma cell type that strongly expresses VEGF, compared to controls, and this was T cell-dependent.

This study is fascinating because VEGF, which is thought to promote tumor growth in solid tumors, functions inhibitory in plasmacytoma.

1. In Figure 3, the authors do not show the effect of IL-2 on tumors in the absence of T cells; tumor growth with the addition of IL-2 alone should be shown.

2. Data on tumor growth in Figure 4 A should be presented.

3. In Figure 4B, there is no data directly showing why it is mentioned in relation to T-cell memory. 

It should be clearly stated how many days after tumor implantation.

4. In Figure 5, immunostaining is relatively less quantitative, so other methods, such as flow cytometry, should be used to assess the increase directly. There is no mention of the increase or decrease or phenotype of lymphocytes in tumors, no functional analysis, and preliminary analysis of antigen specificity and T-cell dependence.

5. Figure 6 shows that T cells express CCR3 in the presence of VEGF. It is unclear why the expression of CCR3 was considered or why there are references to other chemokine receptors. 

6. What about the expression of CCR3 in T cells such as exported lymph nodes in mice showing anti-tumor effects of the study?

Author Response

Comments from the Editors and Reviewers:

Reviewer 2: In this study, tumor growth was suppressed in MOPCSVm, a plasma cell type that strongly expresses VEGF, compared to controls, and this was T cell-dependent.

This study is fascinating because VEGF, which is thought to promote tumor growth in solid tumors, functions inhibitory in plasmacytoma.

  1. In Figure 3, the authors do not show the effect of IL-2 on tumors in the absence of T cells; tumor growth with the addition of IL-2 alone should be shown.

We appreciate this point by Reviewer 2. Though the expression of the IL-2 receptor (IL-2R) and the production of IL-2 by tumor cells have been shown in a variety of malignancies and can regulate tumor growth, the role of IL-2 in tumors is not fully understood [6, 7]. IL-2 has been widely used as a treatment for several tumors based on the finding that this cytokine promotes the proliferation and activity of cytotoxic lymphocytes [1, 2].

It has been widely known that IL-2 doesn’t attack cancer cells directly – it helps the immune system do the job by enhancing the ability of T cells to target and kill cancer cells. IL-2 was initially called T cell growth factor [3, 8]. It stimulates proliferation and enhances function of other T cells and natural killer (NK) cells and B cells.

As shown in Figure 3A, IL-2 addition kills more tumor cells by enhancing the cytotoxicity of CD8 T cells and in Figure 3B, addition of anti-MHC class I antibody to the assay suppresses killing the tumor cells which, we believe, IL-2 itself does not kill the tumor cells directly.

  1. Data on tumor growth in Figure 4 A should be presented.

We agree with Reviewer 2. We have presented the tumor growth in Supplementary Figure 2.

  1. In Figure 4B, there is no data directly showing why it is mentioned in relation to T-cell memory. 

It should be clearly stated how many days after tumor implantation.

We appreciate this suggestion. After 18 days of tumor implantation, the spleens were harvested from the mice bearing MOPCpEF and MOPCSVm. We believe that 18 days after tumor implantation would be reasonable to check whether the memory T cells are formed in our experimental setting according to a previous study [9]. We edited the sentence, which now reads: “non-adherent spleen cells from mice previously challenged with MOPCpEF or MOPCSVm for 18 days were transferred into the lethally irradiated mice.” (Line 282).

  1. In Figure 5, immunostaining is relatively less quantitative, so other methods, such as flow cytometry, should be used to assess the increase directly. There is no mention of the increase or decrease or phenotype of lymphocytes in tumors, no functional analysis, and preliminary analysis of antigen specificity and T-cell dependence.

Consistent with the IHC results in Figure 5, we indeed observed an increase in CD3+ T cells, CD8+ T cells and CD4+ T cells using flow cytometry (Reviewer Only Figure 1). Our future study will be to focus on characterizing the molecular and functional phenotypes of these immune cells within the tumors, including functional analyses, antigen specificity and T cell dependency by treating mice bearing tumors with anti-CD4 and/or anti-CD8 antibodies as suggested by Reviewer 2. These studies are outside of the scope of the current study.

  1. Figure 6 shows that T cells express CCR3 in the presence of VEGF. It is unclear why the expression of CCR3 was considered or why there are references to other chemokine receptors. 

As CCR3 is known to be expressed in T cells and involved in CD4+ and CD8+ T cell trafficking to inflammation and infiltration in the tumors [10, 11], the induction of CCR3 expression by VEGF in T cells stimulated with anti-CD3 and anti-CD28 antibodies suggests a novel mechanism by which tumor-derived VEGF could induce the migration of T cells into a tumor site, in addition to the direct effects of signals immediately downstream of the VEGFR-1 or VEGFR-2 receptors.

Upregulated CCR3 expression suggests that the cells belong to the effector memory T cell population. Besides CCR3 expression, various chemokine receptors, such as CCR5, expression on CD4+ and CD8+ memory T cells indicates a potential to respond to chemotactic gradients and might be important in T cell migration contributing to tumor immune responses in tumor microenvironment [12].

  1. What about the expression of CCR3 in T cells such as exported lymph nodes in mice showing anti-tumor effects of the study?

We appreciate this great suggestion and expect to observe the accumulation of T cells highly expressing CCR3 in both the tumor-draining lymph nodes and at the tumor site. This aspect of the antitumor immunity will be addressed in ongoing future studies and is outside the scope of the current report.

Reviewer 3 Report

The work is potentially interesting

There is a need for numerous changes, corrections of the text and discussion that is not appropriate and adding appropriate references to the appropriate place
1. In the introductory part of the text, line 43, at the end of the sentence add a reference showing the role of cytokines in tumor progression as shown:
Multiomic analysis of cytokines in immuno-oncology. Expert Rev Proteomics. 2020 Sep; 17 (9): 663-674.
2. Regarding the role of VEGF after reference 26, add the reference:
Increased Plasma GDF15 Is Associated with Altered Levels of Soluble VEGF Receptors 1 and 2 in Symptomatic Multiple Myeloma. Acta Haematol. 2021 Nov 24: 1-8.
3. For all primers and other reagents add the country of manufacture as well as the company uniformly. It is written somewhere and not somewhere
4. add references for individual methods
5. In the discussion, add a reference showing the role of immune cells in myeloma for which the authors say there is no data on the association of immune system and myeloma: Decreased CD161 activating and increased CD158a inhibitor receptor expression on NK cells underlies impaired NK cell cytotoxicity in patients with multiple myeloma. J Clin Pathol. 2016 Apr 15: jclinpath-2016-203614.
6) The references used are generally very old and references with more recent data such as:
a) Immune marker changes and risk of multiple myeloma: a nested case-control study using repeated pre-diagnostic blood samples.
Haematologica. 2019 Dec; 104 (12): 2456-2464.
Eb) xtracellular Vesicles Enhance Multiple Myeloma Metastatic Dissemination. Int J Mol Sci. 2019 Jul 1; 20 (13): 3236.
7. Last conclusion: line 407-410 should be reformulated, because it is known that in the advanced stage of the disease there is a decrease in the work of system cells and inefficiency of the immune system to eliminate tumors: Clinical stage-depending decrease in NK cell activity in multiple myeloma patients.
With Oncol. 2007; 24 (3): 312-7.
8) focus the discussion on the basis of the obtained findings
9. In addition, drugs that inhibit VEGF have long existed and new signaling pathway inhibitors are suggested:
a) A novel form of anti-angiogenic molecular antibody drug induces apoptosis in myeloma cells after cultivation upon endothelial feeder cells. Leuk Res. 2021 Sep; 108: 106617.
b) Anti-VEGF Drugs in the Treatment of Multiple Myeloma Patients.
J Clin Med. 2020 Jun 6; 9 (6): 1765.

Author Response

Comments from the Editors and Reviewers:

Reviewer 3: The work is potentially interesting.

There is a need for numerous changes, corrections of the text and discussion that is not appropriate and adding appropriate references to the appropriate place

  1. In the introductory part of the text, line 43, at the end of the sentence add a reference showing the role of cytokines in tumor progression as shown:
    Multiomic analysis of cytokines in immuno-oncology. Expert Rev Proteomics. 2020 Sep; 17 (9): 663-674.

We appreciate this suggestion by Reviewer 3. We have now added this reference in the revised manuscript.

  1. Regarding the role of VEGF after reference 26, add the reference:
    Increased Plasma GDF15 Is Associated with Altered Levels of Soluble VEGF Receptors 1 and 2 in Symptomatic Multiple Myeloma. Acta Haematol. 2021 Nov 24: 1-8.

We have added this reference in the revised manuscript.

  1. For all primers and other reagents add the country of manufacture as well as the company uniformly. It is written somewhere and not somewhere

We have now added this information uniformly in the manuscript.

  1. add references for individual methods

We have now added references for methods used in the manuscript whenever appropriate.

  1. In the discussion, add a reference showing the role of immune cells in myeloma for which the authors say there is no data on the association of immune system and myeloma: Decreased CD161 activating and increased CD158a inhibitor receptor expression on NK cells underlies impaired NK cell cytotoxicity in patients with multiple myeloma. J Clin Pathol. 2016 Apr 15: jclinpath-2016-203614.

We have now added this reference in the revised manuscript.

6) The references used are generally very old and references with more recent data such as:
a) Immune marker changes and risk of multiple myeloma: a nested case-control study using repeated pre-diagnostic blood samples.
Haematologica. 2019 Dec; 104 (12): 2456-2464.
Eb) xtracellular Vesicles Enhance Multiple Myeloma Metastatic Dissemination. Int J Mol Sci. 2019 Jul 1; 20 (13): 3236.

We have now updated references with more recent publications.

  1. Last conclusion: line 407-410 should be reformulated, because it is known that in the advanced stage of the disease there is a decrease in the work of system cells and inefficiency of the immune system to eliminate tumors: Clinical stage-depending decrease in NK cell activity in multiple myeloma patients.
    With Oncol. 2007; 24 (3): 312-7.

We have reformatted content of lines 434-443 to clarify.

8) focus the discussion on the basis of the obtained findings

We appreciate the suggestion. Now the discussion in the revised manuscript is more focused based on the findings.

  1. In addition, drugs that inhibit VEGF have long existed and new signaling pathway inhibitors are suggested:
    a) A novel form of anti-angiogenic molecular antibody drug induces apoptosis in myeloma cells after cultivation upon endothelial feeder cells. Leuk Res. 2021 Sep; 108: 106617.
    b) Anti-VEGF Drugs in the Treatment of Multiple Myeloma Patients.
    J Clin Med. 2020 Jun 6; 9 (6): 1765.

We added these suggestions in the discussion section of the revised manuscript.

Round 2

Reviewer 1 Report

1- Statistical analysis on the experiments is missing and that make it difficult to interpret data

5- The authors did not explain how their pathologist identified MHC-II cells as professional APCs.

Reviewer 2 Report

The analysis related to T cells seems inadequate.
Although it is outside the scope of the claim, a minimum amount of information is necessary when describing T-cell dependence.
We see no room for improvement and cannot accept the results.

Reviewer 3 Report

the authors corrected the manuscript according to most of the instructions suggested